# communications
# earth & environment

# Hydrogen-bearing vesicles in space weathered lunar calcium-phosphates

Katherine D. Burgess [1✉], Brittany A. Cymes [1,2] & Rhonda M. Stroud [1,3]

Water on the surface of the Moon is a potentially vital resource for future lunar bases and longer-range space exploration. Effective use of the resource depends on developing an understanding of where and how within the regolith the water is formed and retained. Solar wind hydrogen, which can form molecular hydrogen, water and/or hydroxyl on the lunar surface, reacts and is retained differently depending on regolith mineral content, thermal history, and other variables. Here we present transmission electron microscopy analyses of Apollo lunar soil 79221 that reveal solar-wind hydrogen concentrated in vesicles as molecular hydrogen in the calcium-phosphates apatite and merrillite. The location of the vesicles in the space weathered grain rims offers a clear link between the vesicle contents and solar wind irradiation, as well as individual grain thermal histories. Hydrogen stored in grain rims is a source for volatiles released in the exosphere during impacts.

[1] Materials Science and Technology Division, U.S. Naval Research Laboratory, Washington, DC 20375, USA. [2] Jacobs, NASA Johnson Space Center, Houston, TX 77058, USA. [3] School of Earth and Space Exploration, Arizona State University, Tempe, AZ 85287, USA. ✉email: kate.burgess@nrl.navy.mil

Spectroscopic observations revealing a widespread hydration signal across substantial portions of the Moon[1–3] have reignited the discussion of the source(s) of lunar water and its mobility on the lunar surface originally raised by the return of Apollo lunar samples[4]. Similarly, although the presence of a tenuous lunar atmosphere, or surface boundary exosphere, was detected and measured by Apollo era experiments[5,6], the first spectroscopic detection of native $H_2$ in the lunar atmosphere was reported using data collected by the Lyman Alpha Mapping Project (LAMP) on the Lunar Reconnaissance Orbiter (LRO)[7,8]. Recent analyses have shown that surface hydration may vary systematically with latitude, temperature, time of day, and presence of magnetic fields[9–14]. Further, telescopic data have shown that molecular water is definitively present in specific locations, potentially trapped within impact glasses or sheltered in voids between grains[15]. $H_2$ does not stick to the surfaces of silicates at lunar equatorial and mid-latitude temperatures[16].

Lunar water is thought to originate from multiple sources, including indigenous reservoirs and external sources such as the solar wind, which implants hydrogen ions ($H^+$). Molecular $H_2$, which can form through reaction between two implanted H atoms (or H and hydroxyl)[17,18] accounts for 7–54% of solar wind protons[19]. However, the mechanism by which solar wind H becomes trapped in the lunar regolith and its speciation has been challenging to interpret from the remote data available[20], and thus the solar wind's contribution to the lunar water budget remains unconstrained.

Laboratory measurements of lunar samples and experimental analogs, as well as modelling, have provided evidence for a relationship between lunar water (including hydroxyl) and the solar wind. Initial results from the Apollo samples demonstrated the release of molecular $H_2$ from some depth within the sample, similar to the profile of solar wind-derived helium[21,22], but these studies were unable to confirm the presence of water. Some agglutinates, complex glass-welded aggregates found in lunar soil, show elevated H as hydroxyl (–OH)[23], and pristine regolith grains with spectral features similar to the remote sensing observations from $M^3$ show a positive correlation between derived $H_2O$ concentration and maturity based on $I_s/FeO$[24]. However, some degree of terrestrial contamination can be difficult to rule out for surface measurements even in pristine samples[25]. Analyses of Chang'E-5 samples have shown the presence of considerable H in grain rims in all phases measured[26]; the abundances are much higher than those measured in Apollo samples and do not suffer from potential exposure to the lunar module cabin atmosphere[25]. However, nanoscale secondary ion mass spectrometry (NanoSIMS) used in some measurements cannot discriminate between H species, and thus studies make the assumption all H is present as –OH[26]. In addition, detailed connection between the hydroxyl or hydrogen measurements and other indicators of space weathering beyond bulk maturity index has been elusive, due to the spatial limitations of Fourier transform infrared spectroscopy and SIMS measurements.

Experiments on lunar soils and analog materials have demonstrated the formation of –OH due to $H^+$ irradiation, and have shown that the –OH signal is stable or slightly decreased at elevated lunar day-like temperatures[27,28]. Others have shown that a combination of ion irradiation followed by heating via pulsed laser to simulate micrometeorite bombardment leads to the formation of water-filled vesicles[29]. In temperature programmed desorption experiments that measure both $H_2O$ and $H_2$, $H_2$ is assumed to be formed from H and –OH combination during the experiment and immediately released[17,30]. Simulations suggest that much of the initially implanted H does react with surface material rather than degassing directly to space as $H_2$[19], but retention and reaction timing are strongly dependent on

parameters that are not well-constrained[18] and could vary greatly based on temperature, composition, and maturity. For example, retention of molecular $D_2$ in irradiated olivine has been shown to rely heavily on the temperature of the sample during ion irradiation[31]. The potential for long-term trapping of molecular hydrogen has not been considered.

The space weathering features of lunar soil particles at the nanoscale provide detailed context to aid understanding of remotely sensed characteristics of the lunar surface[32–37], as well as for potential utilization of resources[32,38]. To that end, we have examined space weathered soil grains of apatite ($Ca_5(PO_4)_3(F,Cl,OH)$) and merrillite (ideal: $Ca_9NaMg(PO_4)_7$), which are the primary reservoir for phosphorous and rare earth elements on the Moon[39]. Apatite is the most common hydrated mineral on the Moon and a common accessory phase on other planetary bodies such as Mercury and multiple asteroids/meteorite types[40–42], and analyzing how it responds to space weathering will aid in understanding how indigenous water sources interact with the solar wind. Both phases analyzed in this study show evidence of volatile-bearing vesicles in their space weathered rims. Our results demonstrate the presence of hydrogen species in these vesicles, and have important implications for the stability and persistence of molecular $H_2$ in regions beyond the lunar poles.

## Results

The apatite and merrillite grains we studied are from mature Apollo lunar sample 79221. The apatite sample is ~6.5 × 1.5 μm with a large portion of the grain surface on multiple sides being available for study (Fig. 1a, b). This apatite grain was identified in the scanning electron microscope (SEM) prior to focused ion beam (FIB) preparation, while the merrillite grain was part of a dirt pile and located only after the sample was in the scanning transmission electron microscope (STEM) (Fig. 1c, d). Space weathering varies between the top and bottom of the apatite, which are defined by how it was mounted in the epoxy and unrelated to its orientation on the lunar surface. The average composition from summed energy dispersive X-ray spectroscopy (EDS) spectra from several maps over the bulk grain indicate the apatite is F-rich (Supplementary Fig. S1, S2). We assume that F +Cl+OH = 1 on a formula unit basis (i.e., (F+Cl+OH):Ca is 1:5); we calculate an equivalent $H_2O$ content of ~1.16 ± 0.09 wt%, within the range of measured mare basalt apatites[39] (Table 1; Supplementary Table S1). The merrillite grain is much smaller and adhered to an agglutinitic glass grain; it has sizeable REE content, as is common in lunar merrillite[43].

**Merrillite**. The merrillite grain is approximately 500 nm across and has 5–6 nm vesicles along portions of its outer edge, particularly near the interface close to the glass (Fig. 2). Compositional analysis with EDS shows minor Y, La, Ce and Nd, which are characteristic of lunar merrillite[43] (Table 1; Supplementary Fig. S2). There are nanophase metallic iron particles ($npFe^0$) throughout the glass to which the merrillite grain is attached, including some concentrated near the boundary between the two phases (Supplementary Fig. S3). Electron energy-loss spectroscopy (EELS) analysis of the low-loss energy range show that the plasmon of merrillite is complex, as expected for Ca-phosphates[44]. However, comparison of spectra from several vesicles in the merrillite rim near the glass show the presence of a peak at ~13 eV that is not present in the average bulk merrillite spectrum (Fig. 2d), indicating they contain H-bearing species. The Ca M-edge in EELS leads to the strong peak in the data at ~35 eV[44]. The low-loss spectra edge shapes and peak height ratios at energies > 20 eV vary across the rim.

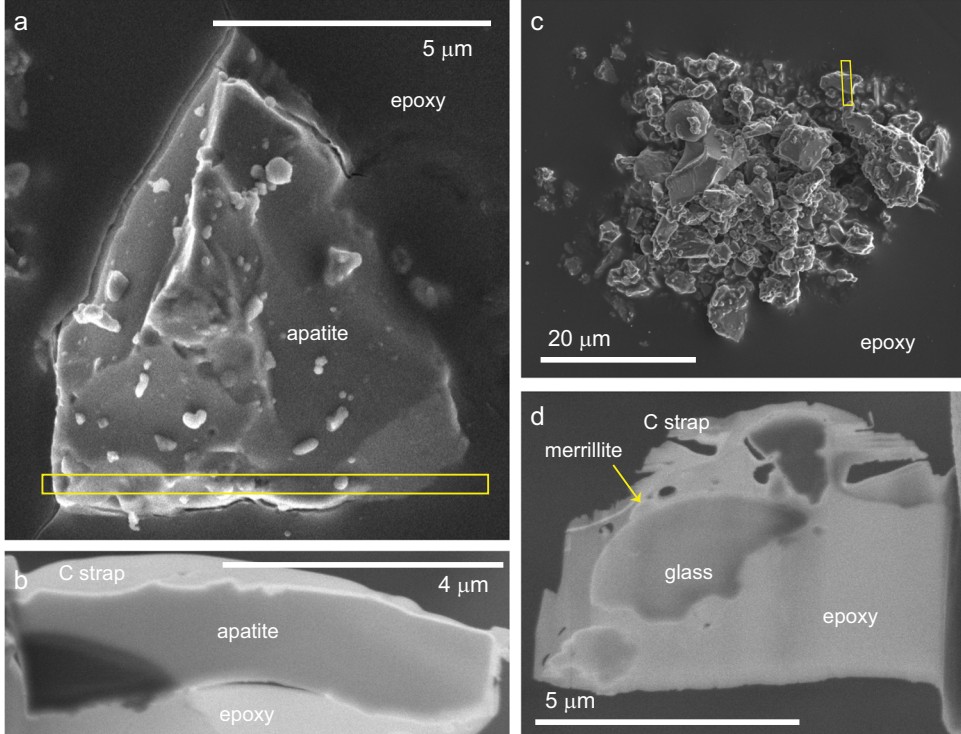

**Fig. 1 Lunar samples analyzed in this study. a** SEM image of apatite particle mounted in epoxy. The sample surface has a number of adhered grains and apparent melt splashes. **b** SEM image of thinned FIB section extracted from location of yellow box in (**a**) showing multiple surfaces of grain available for study of space weathering features. **c** SEM image of dirt pile showing location of extracted slice. **d** SEM image of FIB section that includes a merrillite grain along with several other soil particles.

**Apatite**. The bottom of the apatite (as mounted and oriented in Fig. 1b) shows little evidence of alteration due to space weathering, but portions are coated in vesicular, $npFe^0$-rich silicate glass, which is most likely a melt splash and contains both $Fe^{2+}$ and $Fe^{3+}$ (Supplementary Fig. S4). Most of the top surface of the apatite has a crystalline, vesicular rim with small (2–5 nm) vesicles covered by a thin, poorly crystalline, vesicle-free layer of apatite composition (Fig. 3). The vesicles extend to a depth of ~130 nm. There is a slight decrease in P in the poorly crystalline material at the very surface, possibly caused by solar wind irradiation, but no difference exists in composition between the vesicle-rich layer and deeper material. Several small melt blebs containing Si and O with $npFe^0$ and minor other elements are also present along the surface, coating the poorly crystalline apatite rim (Fig. 3b). Fast Fourier transform (FFT) images from regions along the surface (Fig. 3c insets) demonstrate the poorly crystalline nature of the surface, while the vesicles are present in crystalline material.

Within the vesicular rim, there are several larger, elongate (possibly planar) vesicles (Fig. 4a). Analysis of SAED patterns shows these vesicles lie in the (001) plane when viewed along the [-1 1 0] zone axis (Supplementary Fig. S5). These large vesicles, which are parallel to each other, sit 80-100 nm below the apatite surface, directly beneath a glassy silicate bleb of variable composition that contains $npFe^0$ and nano-sulfides (Fig. 4b). Based on both $t/\lambda$, where $t$ is the thickness and $\lambda$ is the inelastic mean free path of the material, and contrast differences in HAADF, the largest vesicle is about 1/3 the total thickness of the sample, or ~25 nm[45]. The EELS signals for spectra from within the vesicles shows the clear presence of a peak at 13.5 eV that is not present in spectra from pixels directly adjacent to the vesicles (Fig. 4c, d). The sets of spectra, 1 & 2 and 3 & 4, have not been normalized relative to each other. Spectra from spectrum images

acquired first with shorter dwell time per pixel and spectrum images collected after other analyses show similar peak intensity (Supplementary Fig. S6), indicating beam damage does not meaningfully affect this measurement at these conditions. Differences between the spectra inside and outside the vesicles at energies less than 12 eV are below the level of noise. Specifically, we see no clear evidence of a peak near 8 eV, which would be associated with molecular water[46]. Differences at higher energies could be due to the thickness differences caused by the vesicle, or structural or compositional variation around the vesicle. Vesicles that lack a ~ 13 eV peak also display decreased intensity at higher energy relative to adjacent pixels.

Oxygen K-edge EELS shows a pre-peak at ~531 eV that is highly variable in intensity relative to the main edge across the rim (Fig. 4e, f). It is not limited to the vesicles and is broadly present in many areas of the rim. The pre-peak, suggestive of the presence of $O_2$ or of excess O and O-O defects[47–50], is less pronounced in the region directly encompassing the hydrogen-containing vesicle than the surrounding rim material. In some regions of the rim where large vesicles are not present, such as that shown in Fig. 3b, there is no pre-peak, and the shape closely matches that of pure hydroxyapatite standards[51]. The main O-K edge for the apatite has two peaks, at ~537.5 eV and ~540 eV that vary in intensity relative to each other in the region around the vesicles. In general, the 537.5 eV peak is higher where the pre-peak is lower. There is also a clear difference around 545 eV, with an increase in intensity associated with an increase in the pre-peak. These variations do not directly correlate with pre-peak intensity throughout the apatite grain, but could still indicate differences in the relative F, OH, and Cl abundances. These relative peak intensities have been shown to vary between hydroxyapatite and two forms of tricalcium phosphate[51], which have differences in oxygen bonding. The glassy silicate bleb shows

| Table 1 Average composition of Ca-phosphates (wt%). | | |
| --- | --- | --- |
| | **Apatite** | **Merrillite** |
| $SiO_2$ | 2.54 | 0.70 |
| $Al_2O_3$ | 0.56 | 0.42 |
| MgO | 0.20 | 3.40 |
| CaO | 53.27 | 46.85 |
| FeO | 0.66 | 0.86 |
| $Na_2O$ | 0.02 | 0.22 |
| $P_2O_5$ | 36.47 | 40.96 |
| $Y_2O_3$ | 1.35 | 1.88 |
| $La_2O_3$ | 0.15 | 0.46 |
| $Ce_2O_3$ | 0.55 | 2.33 |
| $Nd_2O_3$ | 0.60 | 1.45 |
| Cl | 0.18 | 0.35 |
| F | 2.28 | 0.13 |
| $H_2O$[a] | 1.16 | 0.00 |

[a]Calculated (F+Cl+OH):Ca is 1:5; oxides renormalized.

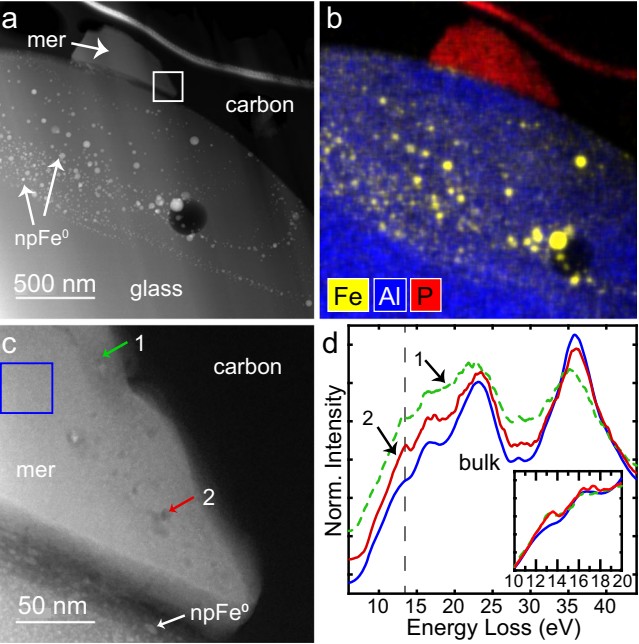

**Fig. 2 Merrillite grain adhered to agglutinic glass. a, b** HAADF image and EDS map of grain adhered to npFe[0]-rich agglutinitic glass. **c** HAADF image from box marked in (**a**) showing the vesicular space weathered rim of the merrillite. The rim of the glass has abundant 1–5 nm npFe[0]. **d** Low-loss EELS spectra extracted from vesicles (1-green dashed line; 2-red solid line) and surrounding material (blue solid line). Several vesicles show clear peaks at 13 eV indicative of the presence of H-bearing species in the vesicles. Spectra are normalized to value at 20 eV and offset vertically for clarity. Inset shows spectra without offset. Mer = merrillite.

a very low pre-peak and a broad main peak centered around 539 eV, consistent with its complex oxygen bonding configurations[52].

## Discussion

The peak at ~13 eV in the low-loss EELS data from vesicles in both the apatite and merrillite grains is a clear indication that they contain hydrogen, most likely as $H_2$. Other molecules, including $H_2O$, $O_2$, and $CO_2$ have peaks at or near 13 eV as well[53,54]. Those molecules, however, include other peaks in both the low-loss and core-loss energy ranges that are not evident in our data. Previous work has shown that hydrated samples will undergo reactions in

the electron beam[46,55,56], including formation of $H_2$. However, in those cases, either other peaks are present, including a peak associated with molecular water at ~8 eV[46], or the interaction of the electron beam with the material caused the formation of the vesicles themselves over time[55]. In the apatite and merrillite samples, the vesicles are present in initial fast-scan images and maintain their shape and size over the course of these measurements. The intensity of the 13 eV peak is also consistent following repeated measurements, indicating we are not inducing the formation of new $H_2$ in the vesicles during our analyses. The lack of lower energy peaks (< 10 eV) show that only $H_2$ is present in the vesicles prior to analysis. Additionally, as noted above, the O-K pre-peak at ~532 eV is broadly present in the rim of the apatite grain rather than being specifically associated with the vesicles, as might be expected if the electron beam was reacting with $H_2O$ or –OH during the analysis to form $H_2$-bearing vesicles.

Based on measurement of the 13 eV peak intensity and the size of the vesicle, we are able to estimate the amount of $H_2$ in the vesicles[57]. For Spectrum 1 in Fig. 4d, we calculate a concentration on the order $10^{27}$ molecules/$m^3$, corresponding to an internal pressure of ~4 MPa at room temperature, assuming ideal gas behavior. A basic estimate of the volume of the vesicle gives a total of 5000–10,000 $H_2$ molecules in the largest vesicle in Fig. 4. Vesicle 2 in the merrillite grain (Fig. 2) has a diameter of ~7 nm, and we calculate the same $H_2$ concentration, within a factor of < 2. The small, round vesicles in the apatite (~3 nm) are < 5% of the total thickness of the sample. If they also have similar concentration of $H_2$, they would contain only tens of molecules, providing constraints on our detection limits at these microscope conditions.

The hydrogen-bearing vesicles in the Ca-phosphate grains are seen only within the space weathered rim, demonstrating the clear link between solar wind irradiation and their formation. Interestingly, the largest vesicles in the apatite, where the largest amount of hydrogen was seen to accumulate, are observed only near surface features indicating likely heating events. In the region shown in Fig. 4, these large vesicles are directly below a silicate melt bleb. The conformation of the silicate glass to the apatite surface indicates that it was molten when it made contact, and thus could have provided considerable heat to the apatite to a depth of several hundred nanometers for up to several seconds based on cooling rates of lunar glasses[58]. Flash heating experiments using lunar soil grains have demonstrated the formation of vesicles after heating to 925 °C for < 1 s[59], while experiments using laser irradiation following ion implantation have linked heating to $OH/H_2O$ formation and additional release of $H_2$ beyond that seen in ion implanted-only samples[29]. Our data, together with these experiments, demonstrate the importance of multiple factors, including high temperature, in the trapping and retention of solar wind hydrogen.

In the apatite, flash heating due to the melt bleb could affect indigenous –OH within the crystal structure in addition to the excess H/OH from the solar wind. Dehydroxylation of hydroxyapatite begins around 800 °C, depending on surrounding water vapor content and apatite composition (i.e., OH, F, and Cl content)[60]. In the decomposition reaction, two –OH groups combine to form one molecule of water and leave excess O within the apatite lattice. However, the conditions on the Moon are highly reducing compared to terrestrial conditions, and solar wind irradiation on even short time scales of tens of years is enough to implant substantial H into the space weathered rim beyond that accommodated by the apatite structure[26]. This could influence the reaction of $H_2O$, $O_2$ and $H_2$, especially at high temperature[61]. As noted here (Supplementary Fig. S4) and in previous work on space weathered lunar samples[36,62,63], oxidized and reduced constituents coexist in these materials.

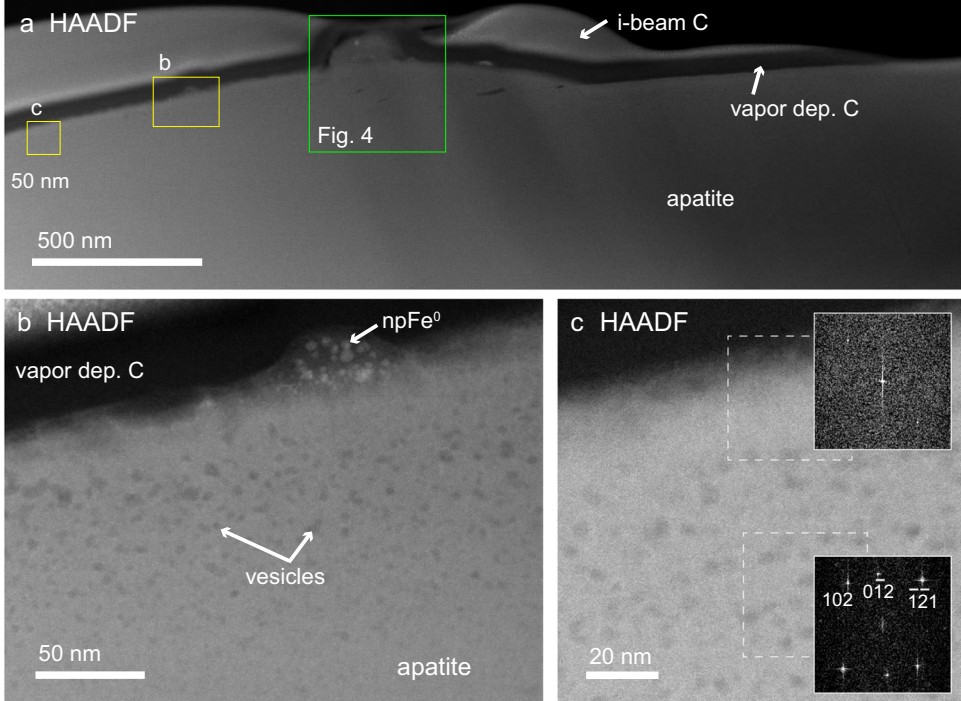

**Fig. 3 Vesicular rim of space weathered apatite. a** STEM HAADF image mosaic of top surface of apatite grain. **b** HAADF image at smaller field of view showing vesicles, vesicle-free rim, and npFe[0]-rich bleb. The vesicles extend to ~130 nm. **c** HAADF image and (inset) FFT patterns showing top ~20 nm is poorly crystalline while region around the vesicles is crystalline apatite. Dashed boxes show locations of FFT patterns.

The variable O-K edge and intense ~532 eV peak can also be related to potential reactions during solar wind irradiation and heating. Simulations of amorphous and crystalline $SiO_2$ have demonstrated that oxygen-excess defects produce a pre-edge peak at ~532 eV[49], as seen in this apatite grain. Molecular $O_2$ has a similar pre-peak and has been identified in vesicles in an interplanetary dust particle[47], although the O or $O_2$ here is not confined to the vesicles. The variation in relative heights of the ~537.5 eV and ~540 eV peaks indicates differences in the apatite structure around the vesicles, consistent with potential dehydroxylation or other changes due to heating. Variability in the shape of the low-loss spectra at energies >20 eV in different locations in the grain rims likely also indicates changes in cation bonding or the crystal structure due solar wind irradiation.

Our identification of a hydrogen signal associated with vesicles in a lunar space weathered rim confirms that a solar wind component is trapped within the grain, and persists in vesicles in detectable amounts even after heating to near maximum lunar daytime temperatures (140 °C). Importantly, we have found trapped molecular $H_2$, which has a short residence time on the sunlit lunar surface of only a few hours[64]. The $H_2$ trapped in vesicles in space weathered rims could provide a reservoir that would be released in pulses by small and large impacts and potentially during microcracking due to diurnal thermal cycling[65]. Vesicles are widespread in lunar space weathered rims of many soil grains across all of the main phases (i.e., plagioclase, olivine, pyroxene, ilmenite)[33–35] and could have real consequences for the timing of volatile release into the exosphere as well as availability of volatile resources by crushing.

Until now, only helium has been unequivocally identified in vesicles in the Apollo samples, primarily in oxides[32] and Fe metal[66]. Our results here show H-species as $H_2$ in phosphate grains, suggesting that diffusion and retention differences between the various lunar phases play an important role in where and how volatiles species are formed, retained, and released on

the lunar surface. Similar to helium-bearing ilmenite[32] and space weathered lunar sulfides[67], some of the vesicles in the apatite appear to be planar and lie in the basal plane of the hexagonal crystal, highlighting the importance of crystal structure in volatile retention. It is also possible that the platelet-shaped vesicles common to hexagonal crystal structures[68] are highly suitable for measurement using STEM when prepared in an advantageous orientation, and thus H or He are more likely to be detected relative to other phases.

Continued measurements of natural samples and experimental simulations across a range of conditions and phases will aid in understanding the variables that affect $H_2$, OH, and $H_2O$ formation and retention related to the solar wind. Recent experiments have shown that the temperature during irradiation can greatly affect the amount of $H_2$ (or $D_2$) released from olivine even after the sample has been stored at room temperature for several days[31]. Irradiation temperature appears to play a large role in natural samples as well, with Chang'E-5 samples, collected from cooler mid-latitude region compared to Apollo sample sites, retaining substantially more H than the more equatorial samples[26]. Studies following the LCROSS mission hypothesized that some $H_2$ released by the impact into a permanently shadowed crater could have been stored within defects in grains[69]. Our current results confirm that $H_2$ can be concentrated in vesicles, similar to He[32], rather than only spread through the rim in individual defects. Given the need for impact heating for formation of water demonstrated by recent experiments[29], it is possible that many small vesicles in lunar space weathered rims are in fact filled with $H_2$.

The role and importance of composition in the long-term trapping of molecular hydrogen in space weathered rims are not well understood. If Ca-phosphates do in fact retain $H_2$ in vesicles more reliably compared to silicates, as suggested by our data, this could influence regional differences in remotely sensed signals of hydration or hydrogen-bearing materials. How or when such

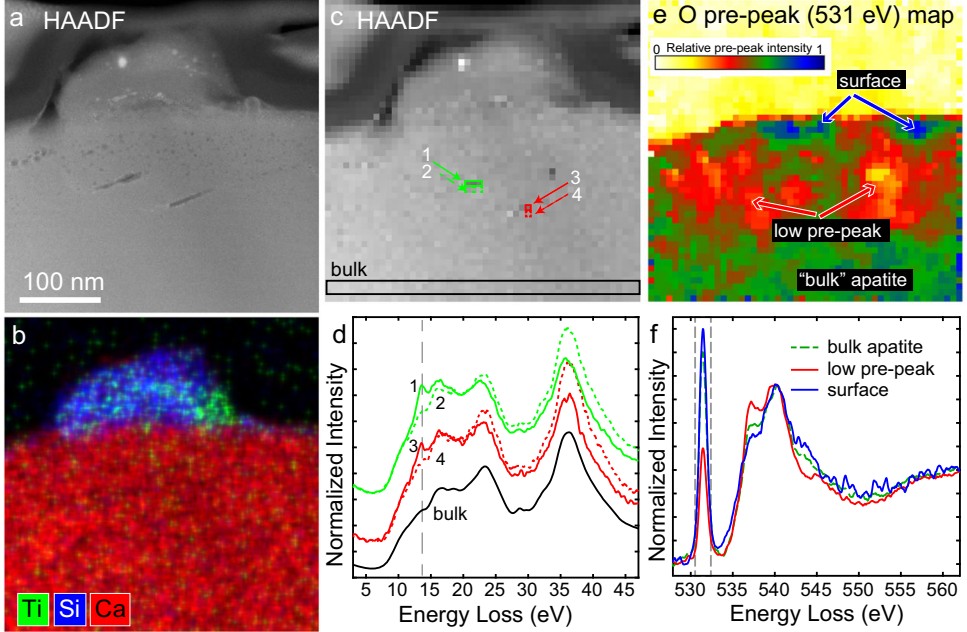

**Fig. 4 Contents of vesicles present in apatite rim. a** HAADF image showing region with melt bleb with large vesicles beneath it. **b** EDS element map shows the bleb is silicate with variable composition of elements including Ti. **c** HAADF image showing individual pixels in the spectra image. **d** Low-loss EELS from pixels noted in (**c**). Solid numbered lines are from pixels within vesicles while dotted lines are from pixels directly beneath each vesicle. Solid black line is from unaltered apatite and shows average low-loss signal for the phase. Spectra from each vesicle offset vertically for clarity. **e** Map of relative intensity of O pre-peak at 531 eV (median filter) showing the regions with smaller pre-peak around the larger vesicles and the surface region with very sharp pre-peak. The silicate and carbon coat have no pre-peak. **f** Selected oxygen core-loss spectra summed from regions with similar pre-peak intensity, normalized at 540 eV. The oxygen K-edge EELS show large variation around the vesicles in this region beneath the silicate melt bleb. Dotted lines indicate window used for pre-peak map.

trapped $H_2$ could be converted to $H_2O$ likely relies on a number of factors that are not well constrained. Apatite and merrillite are minor phases on the lunar surface, comprising up to a few percent in specific lithologies, but often present in much lower amounts. Apatite has an overall estimated ~1% modal abundance[39]. However, as the most common naturally OH-bearing phase, it could contribute meaningful amounts to surface water signals and exospheric cycles. The relative contribution by the indigenous hydration would depend strongly on regolith composition and exposure timing. In regions with low agglutinate content, the apatite and merrillite could contribute an outsized proportion of the water present[70], and some remote data suggest that the KREEP-rich material, which has elevated phosphorous, has a relative increase in surface-bound hydroxyl or molecular water[71]. If the hydrogen content of apatite is increased further due to $H_2$ retention in vesicular rims, the contribution could be further enhanced. Interestingly, this appears to be in contrast to other hydrated minerals affected by space weathering, such as phyllosilicates from asteroid Ryugu, which are dehydrated by the solar wind[72].

The retention of some portion of solar wind hydrogen as $H_2$ in vesicles rather than adsorbed or as part of the structure as either –OH or molecular water has implications for the rates and timing of exospheric cycling of all hydrogen-bearing species. Future work is needed to understand the factors that control the trapping, retention, and speciation, including composition, temperature, and exposure. Additionally, the clear link between space weathering and hydrogen-filled vesicles in apatite and merrillite shows the potential for these minor phases to contribute meaningfully to the total water signal on the lunar surface and its accessibility and mobility. Apatite is a common accessory phase on planetary bodies, including Mercury and several asteroid/meteorite types[40–42]. Therefore, understanding how it is affected

by both the solar wind and micrometeorite impacts, and how those processes work together to form water is very important for understanding volatile cycles on not only the Moon, but many bodies throughout the Solar System.

## Methods

**Sample preparation**. The apatite and merrillite grains are from mature lunar soil 79221 ($I_s/FeO = 81$). The particles were mounted in epoxy such that one surface of the grain was above the surface of the epoxy and available for imaging in the scanning electron microscope (SEM). Focused ion beam (FIB) samples were prepared using a FEI Helios G3 Dual-Beam FIB-SEM equipped with an Oxford 150 mm$^2$ SDD energy dispersive X-ray spectrometer (EDS). Protective straps of C were deposited on regions of interest following imaging, first with the electron beam, then with the Ga$^+$ ion beam. Sections suitable for STEM analysis were extracted using standard approaches at 30 kV and mounted on a Cu half-grid. The final sample thickness in the regions of interest on the apatite was $t/\lambda \sim 0.37$, where $t$ is the thickness and $\lambda$ is the inelastic mean free path of the material, calculated using the log-ratio method.

**STEM with EELS and EDS**. Scanning transmission electron microscopy (STEM) analysis was performed with the Nion UltraSTEM200-X at the U.S. Naval Research Laboratory. Prior to analysis, the sample was held at 140 °C under vacuum for eight hours to drive off adsorbed surface water; a second sample from the apatite grain was loaded after being held under vacuum at room temperature for comparison. The microscope is equipped with a Gatan Enfinium ER electron energy loss spectrometer (EELS) and a windowless, 0.7 sr Bruker SDD-EDS detector. The STEM was operated at 200 kV and ~90 pA, with a 0.1 to 0.2 nm

probe. STEM images were collected in bright field (BF) mode and high-angle annular dark field (HAADF) mode, which is sensitive to atomic number and thickness differences. Maximum pixel time during the scan was 16 µs and care was taken to limit the number of imaging scans needed on regions of interest prior to EELS data acquisition.

EELS data were collected as spectrum images, with a full spectrum over a selected energy range collected per pixel. The energy resolution for EELS, based on the full-width at half-maximum (FWHM) of the zero-loss peak (ZLP) is 0.5 eV. Peak alignment to compensate for systematic energy drift during EELS spectrum image acquisition was carried out using Gatan Digital Micrograph software based on shifts in the ZLP. Background removal from core-loss (O-K and Fe-L) spectra used a power-law fit. Dwell times per pixel are 0.1 ms for spectrum images that include the ZLP and up to 50 ms for core-loss Fe and O. Changes to the material and EELS signal was tracked over multiple scans (Supplementary Fig. S6).

EDS data were also collected as spectrum images, allowing for semi-quantitative mapping of each element of interest, as well as summing of regions with uniform composition for quantification. Compositions were calculated using the Cliff-Lorimer method with instrument-specific k-factors. Oxide wt% is calculated from the cation fractions determined using Bruker Esprit 2.0 software. The sample is thin (i.e., $t/\lambda \ll 1$), so no absorption correction was required. Errors are calculated based on the counting statistics for each summed region.

**Hydrogen concentration**. Low-loss EELS data were used to estimate the amount of gas within vesicles following the method of Walsh et al.[57] developed for He. Gas concentration within each bubble is calculated from

$$n_{H2} = \frac{1}{2} \frac{I_H}{\sigma d I_{ZLP}} \tag{1}$$

where $n_{H2}$ is the number of $H_2$ molecules per $m^3$, $\sigma$ is interaction cross-section of $H_2$, $I_H$ and $I_{ZLP}$ are the intensities of the ~13 eV and zero-loss peaks, respectively, and $d$ is the thickness of the vesicle in the beam direction. The equation has been used for determination of helium content within vesicles in a number of materials[32,57,73,74], and a similar procedure was used by Leapman and Sun[55] for measurement of $H_2$ in bubbles in frozen glycerol. For our experimental conditions, $\sigma = 5 \times 10^{-23}$ $m^2$ [75], and $\lambda = 225$ nm[45]. Sample thickness without vesicles is $t/\lambda \sim 0.35$. The factor of ½ provides the number of hydrogen molecules per unit area rather than hydrogen atoms.

## Data availability
Data were collected using Gatan Digital Micrograph (.dm4) and Bruker Esprit (.bcf) formats. Readers for these data types are available through open-access applications such as HyperSpy. Primary data are available through Zenodo, https://doi.org/10.5281/zenodo.8403583.

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

## Acknowledgements

This work was supported by NASA awards 80HQTR19T0057 (ANGSA) and 80HQTR20T0014 (SSERVI RISE2). The samples were made available by the Lunar Sample Curation Office at NASA Johnson Space Center. This research was performed while Brittany Cymes held an NRC Research Associateship award at the U.S. Naval Research Laboratory.

## Author contributions

K.B.: Conceptualization, investigation, writing – original draft; B.C.: validation, writing – review & editing; R.S.: Resources, supervision, writing – review & editing.

## Competing interests

The authors declare no competing interests.
