## [Peer Review File · Communications Earth & Environment]

14th Jun 23

Dear Dr Burgess,

Please allow us to apologise for the delay in sending a decision on your manuscript titled "Hydrogen-Bearing Vesicles in Space Weathered Lunar Ca-Phosphates". It has now been seen by 2 reviewers, whose comments are appended below. You will see that they find your work of some potential interest. However, they have raised quite substantial concerns that must be addressed. In light of these comments, we cannot accept the manuscript for publication, but would be interested in considering a revised version that fully addresses these serious concerns.

In particular, we will need to see that any revised manuscript meets the following editorial thresholds:

- * Present compelling observational evidence for the presence of hydrogen in association with space weathered vesicles in your samples.
- * Justify your choice of mineral species for analysis and account for the influence of structural hydrogen on your results.
- * Provide sufficient details of your analyses to show that they are robust, including indexing of the crystallographic planes in your fast Fourier Transform images and clarification that your EELS analyses are not adversely effected by beam exposure.

We hope you will find the reviewers' comments useful as you decide how to proceed. Should additional work allow you to address these criticisms, we would be happy to look at a substantially revised manuscript. If you choose to take up this option, please either highlight all changes in the manuscript text file, or provide a list of the changes to the manuscript with your responses to the reviewers.

If the revision process takes significantly longer than three months, we will be happy to reconsider your paper at a later date, as long as nothing similar has been accepted for publication at Communications Earth & Environment or published elsewhere in the meantime.

We understand that due to the current global situation, the time required for revision may be longer than usual. We would appreciate it if you could keep us informed about an estimated timescale for resubmission, to facilitate our planning. Of course, if you are unable to estimate, we are happy to accommodate necessary extensions nevertheless.

Please use the following link to submit your revised manuscript, point-by-point response to the reviewers' comments with a list of your changes to the manuscript text (which should be in a

separate document to any cover letter), a tracked-changes version of the manuscript (as a PDF file) and any completed checklist:

[link redacted]

Please do not hesitate to contact us if you have any questions or would like to discuss the required revisions further. Thank you for the opportunity to review your work.

Best regards,

Joe Aslin

Senior Editor,
Communications Earth & Environment
<https://www.nature.com/commsenv/>
Twitter: @CommsEarth

EDITORIAL POLICIES AND FORMAT

If you decide to resubmit your paper, please ensure that your manuscript complies with our editorial policies and complete and upload the checklist below as a Related Manuscript file type with the revised article:

Editorial Policy Policy requirements (Download the link to your computer as a PDF.)

For your information, you can find some guidance regarding format requirements summarized on the following checklist:(<https://www.nature.com/documents/commsj-phys-style-formatting-checklist-article.pdf>) and formatting guide (<https://www.nature.com/documents/commsj-phys-style-formatting-guide-accept.pdf>).

REVIEWER COMMENTS:

Reviewer #1 (Remarks to the Author):

This manuscript describes the capture of hydrogen from the solar wind in phosphates from Apollo lunar samples. In my opinion, this manuscript could benefit from some clarification.

The positioning of the study could be clarified. The objectives are twofold: volatile formation and water resource potential. While the formation of H₂ seems clear enough in the nano-sized vesicles, it's not clear how they can enrich the exosphere, given that H₂ must surely escape rapidly from the

moon's tenuous atmosphere. As far as water is concerned, the link is even more indirect. In particular, the authors don't really give an estimate of the amount of water that could be generated from the observation of their vesicles (or give a value without justifying it). To avoid increasing the size of their letter, this calculation could be presented in supplementary material.

The choice of mineral phases studied is not really explained. Calcium phosphates were chosen, even though they are only present in very small quantities. Can the observations on these phosphates be extrapolated to other, much more abundant phases? Furthermore, one of these phosphates is already hydrated. In this case, the hydrogen content of these vesicles remains somewhat unclear. Is it the hydrogen implanted by the solar wind, or the structural hydrogen in the apatite?

In my opinion, figure 3 seems a little disconnected from the central theme. It could be moved to supplementary material that would explain the context of these phosphate phases. The "melt splash" of figure 3 and its vesicles are a bit confusing. One also wonders why it's interesting to show this figure in the context of the question posed in this short article.

As far as the merrillite is concerned, it's a small fragment on an impact melt. I would be interested to know whether the surface of the impact melt near this small fragment also contains evidence of space weathering (which it should?).

I have serious reservations about the use of fast fourier transformed images, which can be used to say just about anything if you are not very careful about the conditions under which the images are acquired. The authors show FFT images, which suggests that they have done HR-STEM on this apatite. If so, they should certainly be able to give the orientation of the crystallographic planes that correspond to the elongated vesicles.

In the discussion section, certain parts seem to me to be more like additions to the results and could be moved to the results section. (lines 184-190, lines 191-200)

Lines 289-290. The calculation is not justified. Moreover, it is not specified whether this corresponds to only the irradiated layer or whether it is an average over the whole grain, which is very different.

Finally, I find that the electron beam conditions are not sufficiently described. It is well known that calcium phosphates are sensitive to electrons. At the very least, the current used and the dwell time used for EELS mapping should be given.

Reviewer #2 (Remarks to the Author):

This manuscript reports the detection of H₂ in vesicles in space weathered Ca-apatite grains using electron energy-loss spectroscopy technique. The EELS spectra including hydrogen signals in the low-loss energy range and the O-K edges EELS spectra are well-discussed with references. The clear hydrogen signal in space weathered rim shows the evidence of long retention of H₂, which was thought to be escaped from the sunlit lunar surface within a few hours. To my knowledge, this is the first detection of hydrogen species in space weathered rim in lunar soils by EELS analysis, and the results are significant. The development of the detection methods of solar wind by EELS analysis will influence the planetary science field.

This manuscript reports the detection of H₂ in vesicles in space weathered Ca-apatite grains using electron energy-loss spectroscopy technique. The EELS spectra including hydrogen signals in the low-loss energy range and the O-K edges EELS spectra are well-discussed with references. The clear hydrogen signal in space weathered rim shows the evidence of long retention of H₂, which was thought to be escaped from the sunlit lunar surface within a few hours. To my knowledge, this is the first detection of hydrogen species in space weathered rim in lunar soils by EELS analysis, and the results are significant. I recommend that this paper be accepted after revision.

The major concern of the paper is that the implications for the lunar surface water are ambiguous. The authors emphasized solar wind's contribution to the lunar water budget in the introduction and discussion sections, but an important result in this study might be the detection of H₂, while H₂O originating from solar wind is not detected. The authors may need to clarify whether H₂ found in vesicles is related to the water contents of lunar soils, otherwise, the authors can focus on the importance of H₂ for the lunar exosphere as described in the abstract.

p.6. There is a slight decrease in P...

Is it a result of solar wind irradiation? Could you discuss the cause of the decrease in P ?

p.7. These large vesicles, which are parallel to each other, sit 80-100 nm below the apatite surface..

Are these parallel vesicles related to the crystallographic orientation of apatite? The authors can mention that similar features have been observed in other minerals in lunar soils.

p.8. Previous work has shown that hydrated samples will undergo reactions in the electron beam (Bradley et al., 2014; Jungjohann et al., 2012; Leapman and Sun, 1995), including formation of H₂.

If samples reacted with electron beam, the EELS spectra may have changed over time during the analysis. Did you check the shape of the low-loss EELS during the measurements?

p. 12. Ca-phosphates, which may retain solar wind hydrogen within their structures more reliably than do silicates regardless of the initial OH content.

Do you have any idea why solar wind hydrogen is stored in Ca phosphates, but not in silicate minerals?

p.12. The bulk apatite in this work is estimated to have ~1.2 wt% H₂O, and solar wind irradiation could have increased this, with fully hydroxylated apatite holding up to 3.4 wt% H₂O.

Did the authors assume that bulk apatite could be fully hydroxylated by the effect of solar wind hydrogen, which is concentrated within a depth of only 100 nm from the surface? I understand the importance of the indigenous OH in phosphate minerals as a major contributor to the lunar surface water, but it is not clear whether solar wind irradiation and micrometeorite bombardment on the phosphate minerals significantly affect the water content and behavior on the lunar surface. Please discuss this point in more detail.

Fig. 4c. Please index the FFT patterns.

Fig.5e. Please add a color scale of this figure.

Fig.5f. Explanation about the two dotted lines is necessary.

Response to Reviewers

We appreciate the reviewers' time and comments and have attempted to address the concerns they brought forward. Below, we directly state how we have addressed some of the specific questions. Several of the comments are more general about the focus of the manuscript, and as will be seen in the revised version, significant changes have been made in the introduction and discussion to help focus on H₂ rather than H₂O and clarify how our findings fit within that relationship. We also focus less on the potential ties to indigenous water, at least as regards any attempt at quantification.

Reviewer #1 (Remarks to the Author):

This manuscript describes the capture of hydrogen from the solar wind in phosphates from Apollo lunar samples. In my opinion, this manuscript could benefit from some clarification.

The positioning of the study could be clarified. The objectives are twofold: volatile formation and water resource potential. While the formation of H₂ seems clear enough in the nano-sized vesicles, it's not clear how they can enrich the exosphere, given that H₂ must surely escape rapidly from the moon's tenuous atmosphere. As far as water is concerned, the link is even more indirect. In particular, the authors don't really give an estimate of the amount of water that could be generated from the observation of their vesicles (or give a value without justifying it). To avoid increasing the size of their letter, this calculation could be presented in supplementary material.

As noted above, significant changes have been made to the introduction and discussion to better focus the manuscript on H₂. We have mostly removed attempts to quantify the relationship to water on the surface and in the exosphere, as that is outside of this more focused presentation of our data.

The choice of mineral phases studied is not really explained. Calcium phosphates were chosen, even though they are only present in very small quantities. Can the observations on these phosphates be extrapolated to other, much more abundant phases? Furthermore, one of these phosphates is already hydrated. In this case, the hydrogen content of these vesicles remains somewhat unclear. Is it the hydrogen implanted by the solar wind, or the structural hydrogen in the apatite?

We now focus less on apatite as a source of indigenous water in the exosphere as part of this manuscript, although its potential importance is mentioned. Instead, we have addressed more clearly how the volatile trapping could be related to composition, but if it is not, could have large implications for general presence of H₂ in the lunar regolith.

In my opinion, figure 3 seems a little disconnected from the central theme. It could be moved to supplementary material that would explain the context of these phosphate phases. The "melt splash" of figure 3 and its vesicles are a bit confusing. One also wonders why it's interesting to show this figure in the context of the question posed in this short article.

While we feel the space weathering context of the apatite is important, we agree that the inclusion of Fig. 3 here breaks up the flow of the short article. We have moved Figure 3 to Supplementary material with slight modification to provide this context.

As far as the merrillite is concerned, it's a small fragment on an impact melt. I would be interested to know whether the surface of the impact melt near this small fragment also contains evidence of space weathering (which it should?).

The glass does indeed have space weathering in the form of an npFe⁰-rich rim beneath and next to the merrillite grain. We now note the npFe⁰ in Fig. 2c and have added a figure to the supplementary

material to add further context.

I have serious reservations about the use of fast fourier transformed images, which can be used to say just about anything if you are not very careful about the conditions under which the images are acquired. The authors show FFT images, which suggests that they have done HR-STEM on this apatite. If so, they should certainly be able to give the orientation of the crystallographic planes that correspond to the elongated vesicles.

We have added indexing to the FFT patterns, as well as details of dwell times during the acquisition to the Methods section. Given our imaging conditions and the ways in which damage and changes to the sample did become apparent over time, we are confident in our use of FFT patterns to indicate the thin, mostly amorphous rim on the apatite.

Figure S5 has been added to the supplementary material showing a diffraction pattern that has been indexed to show the alignment of the vesicles along the (001) basal plane of the apatite, which was not apparent from the orientation captured by the FFT pattern.

In the discussion section, certain parts seem to me to be more like additions to the results and could be moved to the results section. (lines 184-190, lines 191-200)

We agree that mention of the sample thickness (previously in 191-200) is more appropriate in the results section and have moved those sentences and adjusted the language accordingly. We also now mention specifically presence or absence of peaks in the EELS data in the results section, but these features are called out specifically in the discussion to support our further points and relationship to previous data.

Lines 289-290. The calculation is not justified. Moreover, it is not specified whether this corresponds to only the irradiated layer or whether it is an average over the whole grain, which is very different.

We assume this refers to the calculation in In 269-270 regarding the amount of water phosphates on the lunar surface could contribute to the exosphere (In 289-290 in our version is the Methods section heading). As noted above, we now make no attempt at this quantification, as we focus more fully on H₂.

Finally, I find that the electron beam conditions are not sufficiently described. It is well known that calcium phosphates are sensitive to electrons. At the very least, the current used and the dwell time used for EELS mapping should be given.

We now include information about the dwell times and current in the Methods. We have also added a sentence in the results referring to additional data in the Supplementary Material showing data from a very fast dwell time and from after a number of other measurements showing that the ~13 eV peak is unchanged.

Reviewer #2 (Remarks to the Author):

This manuscript reports the detection of H₂ in vesicles in space weathered Ca-apatite grains using electron energy-loss spectroscopy technique. The EELS spectra including hydrogen signals in the low-loss energy range and the O-K edges EELS spectra are well-discussed with references. The clear hydrogen signal in space weathered rim shows the evidence of long retention of H₂, which was thought to be escaped from the sunlit lunar surface within a few hours. To my knowledge, this is the first detection of hydrogen species in space weathered rim in lunar soils by EELS analysis, and the results are significant.

The development of the detection methods of solar wind by EELS analysis will influence the planetary science field.

This manuscript reports the detection of H₂ in vesicles in space weathered Ca-apatite grains using electron energy-loss spectroscopy technique. The EELS spectra including hydrogen signals in the lowloss energy range and the O-K edges EELS spectra are well-discussed with references. The clear hydrogen signal in space weathered rim shows the evidence of long retention of H₂, which was thought to be escaped from the sunlit lunar surface within a few hours. To my knowledge, this is the first detection of hydrogen species in space weathered rim in lunar soils by EELS analysis, and the results are significant. I recommend that this paper be accepted after revision.

The major concern of the paper is that the implications for the lunar surface water are ambiguous. The authors emphasized solar wind's contribution to the lunar water budget in the introduction and discussion sections, but an important result in this study might be the detection of H₂, while H₂O originating from solar wind is not detected. The authors may need to clarify whether H₂ found in vesicles is related to the water contents of lunar soils, otherwise, the authors can focus on the importance of H₂ for the lunar exosphere as described in the abstract.

We now focus more on the implications of H₂ specifically. We do discuss briefly the importance of apatite as the main hydrated phase on the lunar surface, but note that any conversion from trapped H₂ to H₂O is not yet understood.

p.6. There is a slight decrease in P...

Is it a result of solar wind irradiation? Could you discuss the cause of the decrease in P ?

We have added "possibly due to solar wind irradiation", which is a reasonable assumption given its loss in the rim only.

p.7. These large vesicles, which are parallel to each other, sit 80-100 nm below the apatite surface..

Are these parallel vesicles related to the crystallographic orientation of apatite? The authors can mention that similar features have been observed in other minerals in lunar soils.

We have added that the vesicles appear to lie in the (001) plane and Figure S5 with SAED patterns. We have also added mention of this aspect to the discussion of the importance of composition or structure in the trapped of volatiles on the Moon.

p.8.Previous work has shown that hydrated samples will undergo reactions in the electron beam (Bradley et al., 2014; Jungjohann et al., 2012; Leapman and Sun, 1995), including formation of H₂.

If samples reacted with electron beam, the EELS spectra may have changed over time during the analysis. Did you check the shape of the low-loss EELS during the measurements?

We have added Figure S6 in the Supplementary Material to help address questions of how the beam could have affected the measurements, which shows very little change to 13 eV peak over a number of scans even as damage to the apatite become apparent in the HAADF images.

p. 12. Ca-phosphates, which may retain solar wind hydrogen within their structures more reliably than do silicates regardless of the initial OH content.

Do you have any idea why solar wind hydrogen is stored in Ca phosphates, but not in silicate minerals?

We speculate that the hexagonal crystal structure common to the apatite and ilmenite could contribute to the retention, but now also note that the relationship could be that these vesicles are just the easiest to measure if they are aligned properly within the STEM given how much of the thickness of the sample they could be.

p.12. The bulk apatite in this work is estimated to have ~1.2 wt% H₂O, and solar wind irradiation could have increased this, with fully hydroxylated apatite holding up to 3.4 wt% H₂O.

Did the authors assume that bulk apatite could be fully hydroxylated by the effect of solar wind hydrogen, which is concentrated within a depth of only 100 nm from the surface? I understand the importance of the indigenous OH in phosphate minerals as a major contributor to the lunar surface water, but it is not clear whether solar wind irradiation and micrometeorite bombardment on the phosphate minerals significantly affect the water content and behavior on the lunar surface. Please discuss this point in more detail.

We have removed discussion of the specific amount of H₂O or OH in this apatite, as it became clear that the mixed focus on H₂ and water led to confusion, and also that any potential contribution to a hydration signal from H₂ being converted is not well understood.

Fig. 4c. Please index the FFT patterns.

Done

Fig.5e. Please add a color scale of this figure.

We have added a bar showing the relative intensity of the peak and noted this in the caption.

Fig.5f. Explanation about the two dotted lines is necessary.

Dotted lines indicate the window used for the O map in 5e (now 4e), which is now noted in the caption.

28th Sep 23

Dear Dr Burgess,

Your manuscript titled "Hydrogen-Bearing Vesicles in Space Weathered Lunar Ca-Phosphates" has now been seen by our reviewers, whose comments appear below. In light of their advice we are delighted to say that we are happy, in principle, to publish a suitably revised version in Communications Earth & Environment under the open access CC BY license (Creative Commons Attribution v4.0 International License).

We therefore invite you to edit your manuscript to comply with our format requirements and to maximise the accessibility and therefore the impact of your work.

EDITORIAL REQUESTS:

*****Please take care to match our formatting and policy requirements. We will check revised manuscript and return manuscripts that do not comply. Such requests will lead to delays. *****

SUBMISSION INFORMATION:

OPEN ACCESS:

Communications Earth & Environment is a fully open access journal. Articles are made freely accessible on publication under a [CC BY license](http://creativecommons.org/licenses/by/4.0) (Creative Commons Attribution 4.0 International License). This license allows maximum dissemination and re-use of open access materials and is preferred by many research funding bodies.

For further information about article processing charges, open access funding, and advice and support from Nature Research, please visit <https://www.nature.com/commsenv/article-processing-charges>

At acceptance, you will be provided with instructions for completing this CC BY license on behalf of all authors. This grants us the necessary permissions to publish your paper. Additionally, you will be

asked to declare that all required third party permissions have been obtained, and to provide billing information in order to pay the article-processing charge (APC).

[link redacted]

Best regards,

Joe Aslin

Senior Editor,
Communications Earth & Environment
<https://www.nature.com/commsenv/>
Twitter: @CommsEarth

REVIEWERS' COMMENTS:

Reviewer #1 (Remarks to the Author):

No additional comments

Reviewer #2 (Remarks to the Author):

The authors focus more on the importance of H₂ in the Introduction and Implications, in response to the reviewer's comments. My questions have almost been addressed with sufficient comments.

Regarding the additional description to address my question about the parallel vesicles:

p. 13. "Similar to helium-bearing ilmenite 32, some of the vesicles in the apatite appear to be planar and lie in the basal plane of the hexagonal crystal, highlighting the importance of crystal structure in volatile retention. It is also possible that the platelet-shaped vesicles common to hexagonal crystal structures 64."

In addition to reference #32, the authors need to refer to the following reference, which shows the appearance of parallel vesicles along the (001) plane in lunar hexagonal iron sulfides by space weathering.

Matsumoto et al. "Space weathering of iron sulfides in the lunar surface environment." *Geochimica et Cosmochimica Acta* 299 (2021): 69-84.

Response to Reviewers:

Reviewer #1 provided no additional comments

Reviewer #2 (Remarks to the Author):

The authors focus more on the importance of H₂ in the Introduction and Implications, in response to the reviewer's comments. My questions have almost been addressed with sufficient comments.

Regarding the additional description to address my question about the parallel vesicles:

p. 13. "Similar to helium-bearing ilmenite 32, some of the vesicles in the apatite appear to be planar and lie in the basal plane of the hexagonal crystal, highlighting the importance of crystal structure in volatile retention. It is also possible that the platelet-shaped vesicles common to hexagonal crystal structures 64."

In addition to reference #32, the authors need to refer to the following reference, which shows the appearance of parallel vesicles along the (001) plane in lunar hexagonal iron sulfides by space weathering.

Matsumoto et al. "Space weathering of iron sulfides in the lunar surface environment." *Geochimica et Cosmochimica Acta* 299 (2021): 69-84.

We have added this reference.